# A thermodynamic investigation of amyloid precursor protein processing by human γ-secretase

Xiaoli Lu[1,2,3] & Jing Huang [2,3✉]

Human γ-secretase cleaves the transmembrane domains (TMDs) of amyloid precursor protein (APP) into pathologically relevant amyloid-β peptides (Aβs). The detailed mechanisms of the unique endoproteolytic cleavage by the presenilin 1 domain (PS1) of γ-secretase are still poorly understood. Herein, we provide thermodynamic insights into how the α-helical APP TMD is processed by γ-secretase and elucidate the specificity of Aβ48/Aβ49 cleavage using unbiased molecular dynamics and bias-exchange metadynamics simulations. The thermodynamic data show that the unwinding of APP TMD is driven by water hydration in the intracellular pocket of PS1, and the scissile bond T32-L33 or L33-V34 of the APP TMD can slide down and up to interact with D257/D385 to achieve endoproteolysis. In the wild-type system, the L33-V34 scissile bond is more easily hijacked by D257/D385 than T32-L33, resulting in higher Aβ49 cleavage, while the T32N mutation on the APP TMD decreases the energy barrier of the sliding of the scissile bonds and increases the hydrogen bond occupancy for Aβ48 cleavage. In summary, the thermodynamic analysis elucidates possible mechanisms of APP TMD processing by PS1, which might facilitate rational drug design targeting γ-secretase.

[1] College of Life Sciences, Zhejiang University, Hangzhou 310058 Zhejiang, China. [2] Key Laboratory of Structural Biology of Zhejiang Province, School of Life Sciences, Westlake University, 18 Shilongshan Road, Hangzhou 310024 Zhejiang, China. [3] Westlake AI Therapeutics Lab, Westlake Laboratory of Life Sciences and Biomedicine, 18 Shilongshan Road, Hangzhou 310024 Zhejiang, China. ✉email: huangjing@westlake.edu.cn

Human γ-secretase is a unique and fascinating four-component protease complex that functions in the final cleavage of the transmembrane domain (TMD) of the 99-aa C-terminal fragment of amyloid precursor protein (APP C99)[1–4]. The subsequent cleavages of APP TMD release amyloidogenic β-amyloid peptides (Aβs) 37-49 amino acids in length, which are closely associated with Alzheimer's disease (AD)[5]. γ-Secretase includes four components, presenilin (PS), presenilin enhancer 2 (PEN-2), anterior pharynx-defective 1 (APH-1), and nicastrin (NCT)[6]. Two catalytic residues (aspartic acids D257 and D385) in subunit PS1 (an isoform of PS) of γ-secretase function to execute endoproteolysis at the ε-site of APP[7]. The aspartic catalytic center at the intracellular pocket of PS1 in γ-secretase can recognize only the β-sheet conformation of substrates (e.g., APP and Notch) instead of their original α-helix conformations[8,9]. Therefore, one interesting question regarding the initial substrate processing is how the α-helical TMD of the substrate unwinds and becomes stabilized by forming a hybrid anti-parallel β sheet between the substrate and PS1.

The unwound APP TMD exposed at the cleavage site interacts with the PS1 catalytic site to form a catalytic core, in which Aβ48/Aβ49 and the corresponding amino-terminal APP intracellular domain (AICD49 − 99 or AICD50 − 99) are first produced via the initial endoproteolysis of γ-secretase. Subsequently, Aβ48/Aβ49 peptides are trimmed every 3 − 4 amino acids through the carboxypeptidase activity of γ-secretase along two main pathways, Aβ49 → Aβ46 → Aβ43 → Aβ40 → Aβ37 and Aβ48 → Aβ45 → Aβ42 → Aβ38[10]. As Aβ40 is a dominant product of wild-type APP, Aβ49 is postulated to be the main product at the ε-site following the first pathway, while Aβ48 production is less probable[11,12]. Recent studies found that Aβ48/Aβ49 selectivity can be altered by mutations in the APP TMD[13–17]. Alexander et al. and Mara Silber et al. suggested that mutants of the APP TMD alter γ-secretase cleavage by modulating the hinge flexibility and conformation of the APP TMD[18,19]. The atomistic understanding of why one species has higher production than the other is still lacking.

Molecular dynamics (MD) simulation is a very powerful tool to study complex conformational dynamics and can reveal the underlying thermodynamics of function-related conformational transitions. Substrate recognition in γ-secretase has recently been investigated with MD simulations by several groups to characterize the corresponding dynamics[20–27]. Han et al. used the coarse-grained (CG) MARTINI force field to perform millisecond-scale MD simulations and demonstrated that TM2/TM6/TM9 of the PS1 domain play an essential role in the initial recruitment of APP TMD[21]. Zacharias et al. and Dominguez et al. uncovered the binding modes of the γ-secretase inhibitors TSA and DAPT in the PS1 domain with MD simulations[22,23], respectively. Wolfe et al. investigated the substrate processing of wild-type and mutant APP by γ-secretase using Gaussian accelerated MD simulations and found thermodynamic evidence that mutations in APP can shift its active conformation, thus altering Aβ48/Aβ49 product ratios[24]. Chen et al. performed free energy simulations on the wild-type protein and familial Alzheimer's disease (FAD) mutants to reveal the free energy difference between hydrogen-bonded and water-bridged states, explaining the correlation between FAD mutations and experimental activities[25]. All-atom and CG simulations have also been used to investigate the flexibility of γ-secretase among various substrate-bound states[27]. The focus has been mostly on the conformation of γ-secretase and the interaction interface by assuming that a substrate is bound;[28] however, how the endogenous substrate APP TMD responds to γ-secretase has raised less attention and has not been studied systematically.

In this work, we investigated the essential steps of APP substrate processing by γ-secretase using all-atom unbiased MD and bias-exchange metadynamics (BE-MetaD) simulations[29]. First, assuming an APP substrate has been recruited by γ-secretase and enters PS1, we studied how the α-helix of APP TMD unwinds and forms an anti-parallel β sheet when it binds and is processed by PS1. The free energy profiles constructed along with the APP TMD unwinding and binding processes provide thermodynamic details on the stepwise APP TMD unwinding process and the coupling between the conformational changes and PS1 binding. Next, we studied the correlation between conformational fluctuations in the catalytic core and the Aβ48/Aβ49 production ratio to interpret the higher Aβ49 production from the wild-type APP TMD. Simulations of APP TMD with the FAD mutation T32N were also carried out to explain how the Aβ42/Aβ40 production ratio can be altered by interference with conformational dynamics. Together, our calculations generate useful insights into the mechanism of APP TMD recognition by γ-secretase and provide a thermodynamic perspective for subsequent conformational changes and specific Aβ production.

## Results and Discussion

**PS1 creates a hydrophilic environment for the spontaneous unwinding of APP TMD.** To investigate the thermodynamics of APP unwinding, we set up simulation systems using an α-helical APP TMD freely solvated in water and embedded in membrane (PDB id: 2LlM)[30], as well as encountered in the PS1 domain extracted from the cryo-EM complex structure (PDB id: 6IYC)[8] (Fig. 1). We carried out BE-MetaD simulations with three biased replicas to drive the unwinding process and one neutral replica, and each replica was run for 100 ns. The residual helicity of 12 C-terminal residues (VIVITLVMLKKK) of APP TMD ($f_H^{C12}$) is the percentage of the content of ideal α-helix block in a peptide, which ranges from 0 (fully unwound) to 1 (all in α-helix conformation) and was used as a collective variable (CV) to characterize the unwinding state of APP TMD.

The unwinding free energy profiles as a function of $f_H^{C12}$ are shown in Fig. 2a. The APP TMD adopted a stable helical conformation ($f_H^{C12}$ of 0.8 ∼ 0.9) in the membrane environment but fully unwound in water, as only one minimum ($f_H^{C12}$ ∼ 0.1) was observed. This demonstrates how the secondary structure of a protein or peptide can be modulated by its environment. In the hydrophobic environment of lipid bilayers, the TMD forms an α-helix to maximize the number of hydrogen bonds between the backbone amide and carbonyl groups, while in water, hydrogen bonds can be formed between water and the TMD such that the enthalpy gain is outweighed by the entropy penalty from the regular secondary structure[31]. Interestingly, two stable conformational states were identified for the PS1-bound APP TMD, which respectively exhibit similar $f_H^{C12}$ and curvature to those in the membrane and the water environment. The local minimum corresponding to the α-helix ($f_H^{C12}$ of 0.8∼0.9) is 3.9 kcal/mol higher than the global minimum located at $f_H^{C12}$ ∼ 0.1, with an activation barrier of 5.4 kcal/mol, suggesting that the APP TMD unwinds spontaneously in the intracellular pocket of PS1.

The free energy profiles obtained from BE-MetaD simulations are consistent with the observations from 200-ns unbiased MD simulations of the APP TMD in different environments (Supplementary Fig. 1, Supplementary Data 1). In water, the $f_H^{C12}$ of the APP TMD fluctuates significantly, while in the membrane, the helix remained very stable. An unwinding tendency occurred in the 200 ns unbiased MD simulation when it was inserted into the PS1 environment (Supplementary Fig. 1a–d), although full unwinding was not observed due to the limited simulation time scale. The hydrophilic interface of PS1 enhanced water accessibility (Supplementary Fig. 1f) and helix dynamics compared to those in pure lipid bilayers.

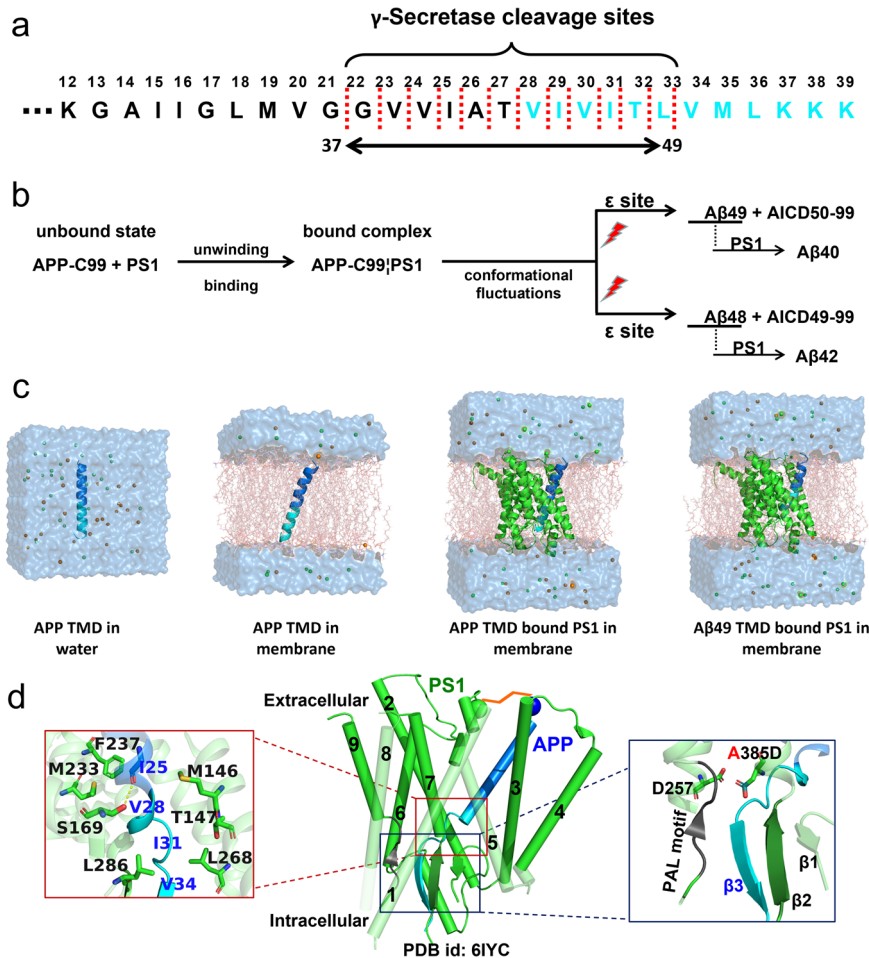

**Fig. 1 Overview of MD simulation systems. a** Sequence overview of Aβ49 APP TMD and successive cleavage sites for Aβ37-49 production, the 12 C-terminal residues of APP TMD are highlighted in cyan. **b** The proposed process of APP C99 cleavage by γ-secretase for the production of a series of Aβs. **c** Simulation systems were used to investigate the dynamic and thermodynamic properties of APP TMD (in blue) recognized by PS1 (in green) of γ-secretase, including APP TMD solvated in the water environment, APP TMD embedded in the membrane environment, APP TMD bound to PS1, and Aβ49 TMD bound to PS1. **d** Enzyme-substrate interaction between APP TMD and PS1 of γ-secretase with key residues of PS1 involved in the interaction with unwound APP highlighted. The PAL motif (P433-A434-L435) and two β sheets (β1 and β2) from PS1 stabilize the APP β sheet via the hybrid anti-β sheet formation. The mutation A385 for the experimental structure determination is restored as D385 in the simulation systems.

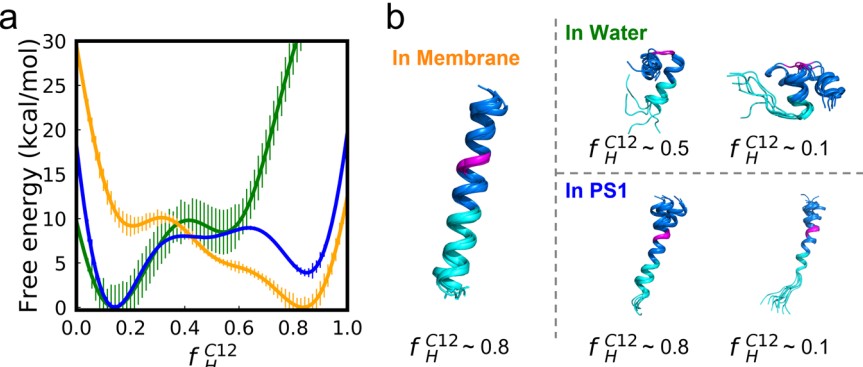

**Fig. 2 The hydrophilic environment provided by PS1 favors the spontaneous unwinding of APP TMD. a** Free energy profiles of unwinding as a function of residual helicity for the C12 region of the APP TMD, corresponding to APP TMD in membrane (orange), in water (green), and in the PS1 domain (blue). Uncertainty is estimated with block analysis. **b** Structural bundles of APP TMD in three simulation systems superimposed onto the middle of the APP TMD. Cyan: C12 region, last 12 residues on the C-terminal helix of APP TMD; magenta: Gly-Gly hinge in APP TMD. $f_H^{C12}$: the residual helicity of the C12 region.

From structural clustering of APP TMD in three environments, we note that the Gly-Gly hinge in the middle of APP TMD is disturbed and undergoes obvious twists and turns in aqueous environments but is constrained when bound to PS1. The unwinding of the C12 region occurs stepwise to extend the helix in the pure water environment and the hydrophilic interface provided by PS1 (Fig. 2b, $f_H^{C12}$~0.8, ~0.5, ~0.1), where several polar side chains are located, including Q276, N279, E280, T281, L286, S289, and K380 (Supplementary Fig. 1f). The hydrogen bonds broken per turn of the α-helix are successively driven by the surrounding water molecules, facilitating the exposure of the scissile bonds in the catalytic core of PS1. The deepest position to which water molecules can penetrate into PS1 is near the V28-I29-V30 region of the APP TMD. We note that the first cleavage site producing Aβ49/Aβ48 is the T32-L33-V34 region, where the hydrogen bonds in V28/T32, I29/L33, and V30/V34 are disrupted in a hydrophilic environment (Supplementary Fig. 1f). By measuring the H/D exchange rates with NMR spectroscopy, Cao et al. found that the strengths of some backbone hydrogen bonds in the APP TMD are weaker than those of water–water hydrogen bonds. Interestingly, those weak backbone hydrogen bonds are at or near preferred γ-secretase cleavage sites[32].

**Binding of unwound APP TMD to the PS1 recognition surface promotes entering into the lowest energy states.** The unwinding of the APP TMD allows it to form stable contacts with γ-secretase for further processing. To understand the thermodynamics of coupled unfolding and binding, we constructed the 2D free energy profile as a function of $f_H^{C12}$ and the coordination number (CN) between the C12 region and PS1 from BE-MetaD simulations with three CVs to describe the helix unwinding of APP TMD and another three CVs to describe the binding between its C12 region and PS1 (see computational methods). As evident in Fig. 3a, the unwinding and binding processes are strongly coupled and occur stepwise. When the TMD is α-helical (S0), few contacts can be made with PS1 residues (CN < 40), and the C12 region may partially unwind at its terminal LKKK- residues ($f_H^{C12}$ of 0.7~0.8). Further unwinding is accompanied by a significant increase in the contacts between the APP TMD and PS1 and the establishment of additional hydrogen bonds (S1→S2→S3→S4). In those conformational states, the C12 region of the APP TMD is fully unwound in states S3 and S4. S3 is the global minimum ($f_H^{C12}$ of 0.1~0.2, CN = 135) and is 7.5 kcal/mol more stable than S0. States S1 and S2 have similar $f_H^{C12}$ (0.4~0.6) but can be differentiated by the degree of binding with PS1. The 2D free energy landscape is qualitatively consistent with but quantitatively

distinct from the 1D helix unwinding profile (blue curve in Fig. 2a), which suggests that binding is a coupled slow dynamics mode and demonstrates how the binding of the APP TMD with PS1 further stabilizes its unwound states.

The multiple intermediate binding states of the APP TMD bound to PS1 were further inspected at the atomic level to reveal the roles that certain interactions played in the coupled unwinding and binding process. Cluster analysis was performed on these successive intermediates to understand the atomic details of the sampled conformations. In particular, the interaction contributed by the GVKLG motif (located in the loop between TM6 and TM7) and the PAL motif (located in the loop between TM8 and TM9) of PS1 was analyzed. These two motifs were known to be involved in the substrate recognition of γ-secretase for the active conformation[33], as their mutations or deletions can directly alter or even eliminate the substrate cleavage activity[34,35]. In state S4, interactions with unwound APP TMD mainly involved the GVKLG motif at the intracellular pocket interface and the PAL motif between TM8 and TM9 of PS1, which is consistent with the cryo-EM structures and experiments showing that perturbing those two regions can alter substrate recognition[36,37]. In states S1 and S2, the GVKLG motif gradually forms hydrogen bonds with the stepwise unwinding of the TLVMLKKK region (Supplementary Fig. 2a). Comparing states S3 and S4, we found that S4 is very similar to the cryo-EM structures determined by Zhou et al.[8], instead of the global minimum S3. The cross-linking strategy and D385A mutation used in the experiment to overcome the extremely transient nature of the complex is a possible reason they obtained an ensemble-average structure of APP recognized by γ-secretase[8]. The artificially covalent attachment of the substrate might introduce non-native contacts. The difference might also be due to the 'cryogenic' condition in the cryo-EM experiments, as the biological entities examined are frozen in vitreous ice[38]. A recent simulation study on γ-secretase showed that cryo-temperature can affect the radius of gyration and the β-sheet formation of substrate-bound γ-secretase[39].

We further assessed the stability of the binding between APP and PS1 in state S4 with conventional MD simulations. In the 200 ns MD simulation, the binding pose between the two PS1 motifs and unwound APP TMD transitioned from S4 to S3. As shown in Fig. 3b, the GVKLG motif stably interacts with the LVMKL region in the APP TMD by forming an anti-parallel β sheet in both S3 and S4, while the loop containing the PAL motif undergoes a significant conformational shift in S4. From the structural comparison of the APP-bound state (PDB id: 6IYC) and the apo state (PDB id: 5A63)[40] of PS1, we found that the

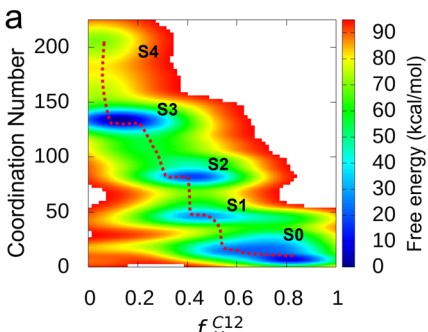
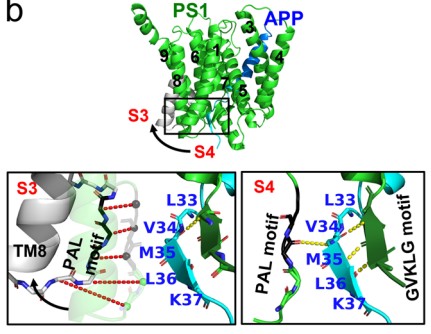

**Fig. 3 Coupling between the unwinding of APP TMD and its binding with the PS1 intracellular interface. a** Free energy profiles as a function of residual helicity for the C12 region of the APP TMD and the coordination number between PS1 and the APP TMD. All the intermediate states in the APP recognition process are labeled S0, S1, S2, S3, and S4. **b** Structural comparison of the GVKLG motif and PAL motif binding to the unwound part (cyan) of the APP TMD (blue) in states S3 and S4 (the PAL motif is in black, and the GVKLG motif is a forest-green β sheet). The red dashed lines represent the distance at which the Cα atom at the PAL-containing loop moves with the conformational shift.

structural difference occurs mainly at the loop between TM8 and TM9 (Supplementary Fig. 2b), indicating the high flexibility of this PAL-containing loop. The occupancy of the anti-parallel β sheet formation in unbiased MD simulations also illuminates the dynamic difference of the two motifs (Supplementary Fig. 2c). The dynamic properties of the two key motifs suggest that the GVKLG motif in PS1 likely anchors the unwound APP TMD via a typical anti-parallel β sheet, while the PAL motif serves as a switch to shift the cleavage site closer to the catalytic sites for subsequent proteolysis.

**Linking the structural configurations of the APP-PS1 complex to APP activation and cleavage.** We next investigated the preferred structural configurations of the catalytic core for Aβ48/Aβ49 production. After helix unwinding, the first Aβ cleavage site located at the T32-L33-V34 region of the APP TMD is exposed and delivered close to the catalytic sites D257/D385 of γ-secretase. Essentially, the cleavage process requires conformational changes that position the substrate precisely for subsequent enzyme-substrate recognition to form the Michaelis complex for proteolysis[41]. A generally accepted mechanism of aspartic proteases starts with the nucleophilic attack of an activated water molecule on the carbonyl C atom of the scissile bond and undergoes nitrogen inversion with a concerted rearrangement of electron pairs and proton transfer to the amide NH, resulting in cleavage of the peptide bond and product release[42]. It is commonly assumed that the cleavage of APP occurs every three amino acids along either the Aβ48 → Aβ45 → Aβ42 or Aβ49 → Aβ46 → Aβ43 → Aβ40 pathways, leading to the major final products Aβ42 and Aβ40, respectively.

Using the unwound APP-bound complex (state S3) as the starting structure, we sampled the stable configurations of the catalytic sites in 200 ns unbiased MD simulations with three replicates and measured the Cγ-Cγ distances between D257 and D385 (Fig. 4a). When the active site was in a monoprotonated state, both hydrogen-bonded and water-bridged D257/D385 were captured (Fig. 4b), and the Cγ-Cγ distances were approximately 4.5 ± 0.4 Å (orange curve) and 6.5 ± 0.3 Å (green curve), respectively. The time evolution of the distance between the O atom of water and the center of mass (COM) of the Cγ atom of D257/D385 indicates that water molecules are dynamically exchanged in the water bridge (Supplementary Fig. 3). When both aspartic acids were unprotonated, the conformation of the APP-PS1 complex was inactive for Aβ cleavage (Fig. 4b), as the Cγ-Cγ distance was larger than 8 Å (Fig. 4a, blue curve); thus, the scissile bonds of the T32-L33-V34 region on APP were distant from the side chains of D257/D385 (Fig. 5a).

Then, we investigated the coordination between the scissile bonds of T32-L33-V34 on APP and the side chains of D257/D385. As shown in Fig. 5b, we sampled a major configuration where the backbone NH group of V34 on APP donates a hydrogen bond to the carboxyl group of D385 for Aβ49 cleavage. This major configuration is attributed to the coordination between the L33-V34 scissile bond on APP and side chains of D257/D385 (Fig. 5c), while slippage of the T32-L33 scissile bond into the valley of D257/D385 for Aβ48 cleavage was not observed. To explore the conformational change that allows Aβ48 cleavage, more advanced enhanced sampling and free energy simulations are needed.

**Conformational fluctuations of the catalytic core alter Aβ48/Aβ49 cleavage.** We further investigated which scissile bond (T32-L33 or L33-V34) is subjected to catalysis by D257/D385 for Aβ48/Aβ49 cleavage from a thermodynamic perspective with BE-MetaD simulations. Notably, we speculated that stable structural configurations are established before the cleavage reaction. The 2D free energy profiles as a function of the coordination number for Aβ48 (cleaving T32-L33) and Aβ49 (cleaving L33-V34) production were constructed for different D257/D385 protonation states. At high pH with both D257 and D385 unprotonated, the catalytic core is inactive, as the global minimum C0 state located at the origin has little coordination between the catalytic site and either scissile bond of the APP TMD (Fig. 6a).

With one aspartic acid protonated (D257), the global minimum transitions to an active conformation for Aβ49 cleavage, where the side chains of D257/D385 directly access the L33-V34 scissile bond (Fig. 6b). It is worth noting that there is a local minimum (C1) close to the global minimum (C2). We found that the minimum C1 matches the stable configuration in Fig. 5b, where D385 accepts a hydrogen bond from the L33-V34 scissile bond that directly hydrogen bonds with D257. In contrast, in state C2, D257 and D385 are water-bridged (Fig. 6b). The free energy difference between C1 and C2 is 1.92 kcal/mol, indicating that the water-bridged state C2 is more favorable. This result is consistent with the free energy profile along the Cγ-Cγ distance of D257/D385 estimated by Chen et al., where the free energy difference between the hydrogen-bonded and water-bridged states is 0.62 kcal/mol without considering coordination with the scissile bonds of T32-L33-V34[25].

Another local minimum C3, where the T32-L33 scissile bond shifts into the catalytic sites to allow Aβ48 cleavage, was found to be separated from C2 by a free energy difference greater than 6.4 kcal/mol. Similar observations were made in the free energy profile built with protonated D385 and unprotonated D257

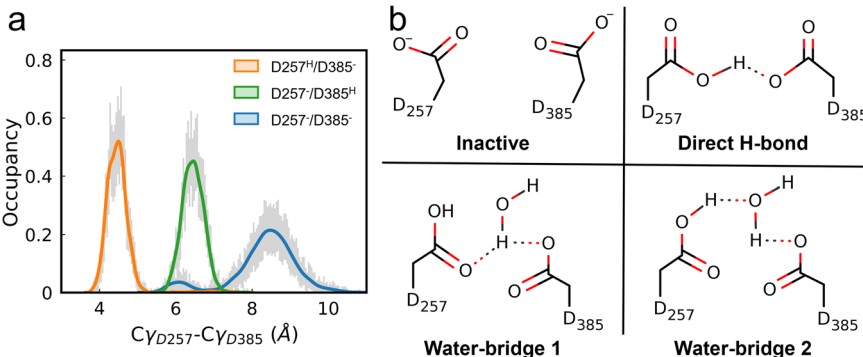

**Fig. 4 The activation of catalytic core is dependent on the protonated states of D257 and D385. a** The distribution of the Cγ-Cγ distance between catalytic sites D257 and D385 in unbiased MD simulations. Uncertainty is estimated with histogram analysis. **b** The stable configurations of D257/D385 are identified according to the Cγ-Cγ distance, including an inactive unprotonated state and three monoprotonated states where the side chains of the two aspartic acid residues are either directly hydrogen-bonded or water-bridged.

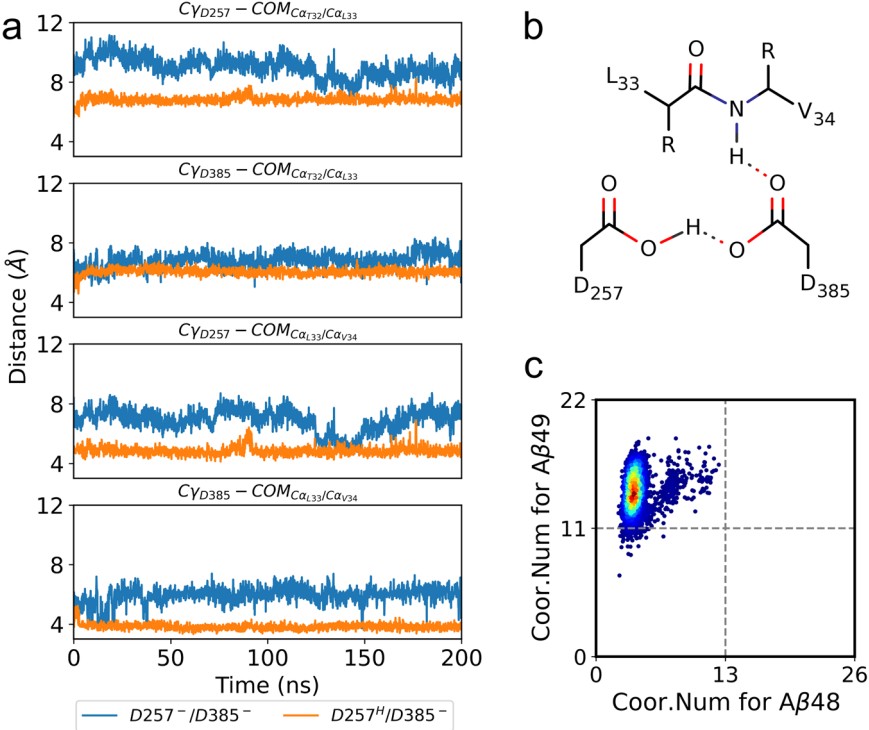

**Fig. 5 The configurations of catalytic core are altered by the protonated states of D257 and D385. a** Time evolution of the distance between the Cγ atom of D257/D385 and the COM of the Cα atom in T32-L33 or L33-V34 under two protonation states. **b** The stable configurations of the catalytic core for Aβ49 production. **c** The formation of coordination between PS1 and scissile bonds in T32-L33 (for Aβ48 cleavage) or L33-V34 (for Aβ49 cleavage) at the catalytic core.

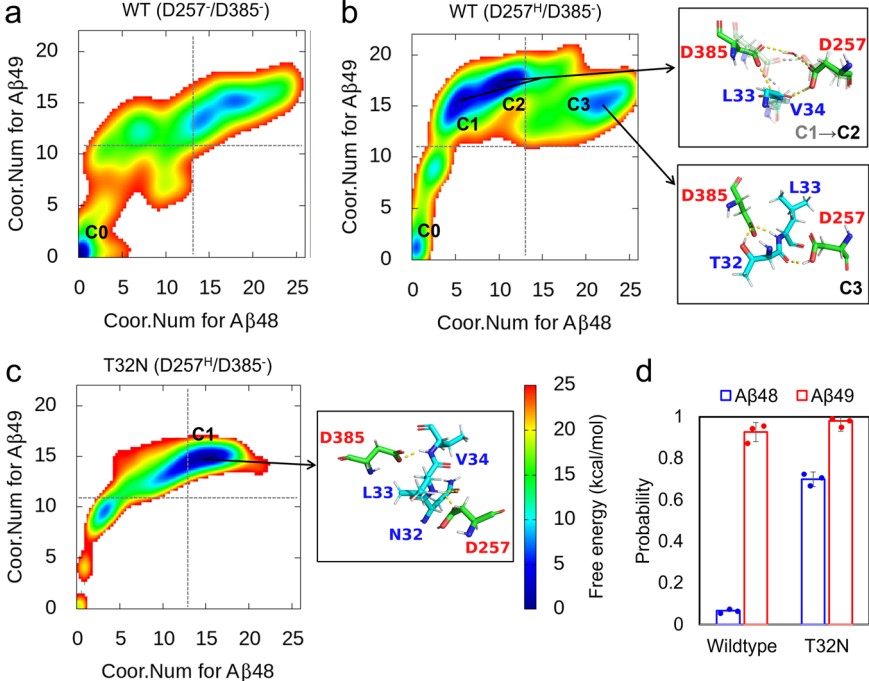

**Fig. 6 Structural configurations formed in the wild-type system prefers Aβ49 cleavage while the T32N mutation on APP alters the Aβ48/Aβ49 cleavage ratio. a–c** 2D free energy profiles of coordination formation for Aβ48 and Aβ49 cleavage from BE-MetaD simulations, as well as representative conformations corresponding to free energy minimums. **a** Inactive wild-type protein without protonation (D257⁻/D385⁻), (**b**) active wild-type protein with protonated D257 and unprotonated D385 (D257^H/D385⁻), (**c**) active T32N mutant with D257^H/D385⁻. (**d**) The occupancy of the hydrogen bonds between scissile bonds T32-L33 or L33-V34 and D257/D385 from the unbiased MD simulation for the wild-type and T32N mutant APP systems. Data are presented as mean values +/− standard deviation.

(Supplementary Fig. 4), where the local minimum for Aβ48 cleavage is 3.8 kcal/mol, which is smaller than the global minimum for Aβ49 cleavage. These results show that as long as D257/D385 is monoprotonated, the APP-PS1 complex remains active for subsequent proteolysis. Moreover, the lower free energy associated with stable coordination with the L33-V34 scissile bond explains why the production of Aβ49 is higher than that of Aβ48 (thus Aβ40 is higher than Aβ42) in wild-type APP TMD processed by γ-secretase. The occupancy of hydrogen bonds that D257/D385 forms with the L33-V34 scissile bonds is approximately 10 times higher than that with T32-L33 in wild-type APP (Fig. 6d). Our simulation results are qualitatively consistent with the widely reported 6- to 9-fold production ratio of Aβ40/Aβ42 in the wild-type APP-PS1 system[43].

We further considered the effect of FAD mutation on Aβ's cleavage alteration and built a simulation system with the T32N mutant of APP for comparison with the wild-type APP system. This FAD mutation is located at the cleavage site for the initial Aβ48 cleavage and results in an increased production ratio of Aβ42/Aβ40[16]. As shown in Fig. 6c, the free energy landscape of catalytic complex formation was significantly altered by the mutation. The C1, C2 and C3 minimums in the active wild-type free energy profile vanish, and only one global minimum where the side chains of D257 and D385 form two hydrogen bonds with the scissile bonds T32-L33 and L33-V34, respectively, is identified (Fig. 6c). The hydrogen bond occupancy for Aβ48 cleavage increased from $0.07 \pm 0.01$ to $0.71 \pm 0.06$ in the mutant APP system (Fig. 6d). Both the thermodynamic properties and structural dynamics support that the T32N mutation likely increases the Aβ48/Aβ49 cleavage ratio, which is consistent with the increased Aβ42/Aβ40 production ratio estimated by Karch et al.[17].

**Successive delivery of the cleavage sites on the APP TMD into the catalytic site of PS1.** Assuming that the proteolytic reaction occurs at the L33-V34 scissile bond of the APP TMD, the produced Aβ49 will deliver the successive cleavage sites into the catalytic site of PS1. In terms of substrate recognition, the delivery mechanism following endoproteolytic cleavage at the ε-cut site is still an open question[28]. Yang et al. proposed two possible delivery mechanisms for tandem cleavage in APP-PS1, i.e., the unwinding model and piston model, and speculated that the former is more favorable based on the complex structure and the properties of the substrate TMD sequence[9]. We have demonstrated that unwound APP TMD in the intracellular pocket of PS1 is the most energetically stable conformation (Fig. 2a, blue line). Here, we further explored whether the neighboring helical portion in the APP TMD for tandem cleavage also exhibits similar thermodynamic properties. A simulation system including an Aβ49 TMD bound to PS1 was set up for free energy calculations with BE-MetaD simulations (see computational methods).

We constructed the 1D and 2D free energy profiles as a function of $f_H^{C12}$ for the C12 region of the Aβ49 TMD (-GVVIATVIVITL), as well as a function of the above value and the coordination number between PS1 and the Aβ49 TMD, respectively (Fig. 7). Regardless of the binding to PS1, a minimum located at $f_H^{C12} \sim 0.1$ indicates that the unwound Aβ49 TMD also reaches a stable state (Fig. 7a). This result indicates that PS1 also provides a favorable environment for the unwinding of the neighboring α-helix, which is consistent with the fact that γ-secretase can recognize substrates with little sequence specificity in the TMDs[44]. From the 2D free energy profile in Fig. 7b, we found two minimums in the free energy landscape, states S0' and S1'. The global minimum (S0') is where the Aβ49 TMD remains α-helical and rarely makes contact with the intracellular interface

of PS1. Consistent with the 1D results, the cleavage-preparing state S1' is not the global minimum in the 2D free energy profile. In the local minimum (state S1'), a new anti-parallel β sheet forms between the GVKLG motif and the -IVIT- region of the Aβ49 TMD, and the A26-T27 scissile bond is released from the unwound Aβ49 TMD to interact with catalytic sites (Fig. 7b). Our simulations provide thermodynamic evidence that the unwinding of the substrate APP can deliver successive cleavage sites into the active site of PS1. However, other driving forces, such as conformational changes not fully captured in our simulations (e.g., bending and twisting of the Aβ49 TMD), might play additional roles in further stabilizing the state for subsequent cleavage.

## Conclusions

In this work, we carried out atomistic simulations to provide insight into the mutual conformational adaptation of the substrate-enzyme complex during exosite contacts and induced-fit unfolding. Given the large structural change during unfolding, the transition state energy in the last step is expected to be high and is most likely the rate-limiting step. Assuming that an APP substrate recruited by γ-secretase enters PS1, we uncovered the molecular mechanisms of APP TMD adaptation to the intracellular interface of PS1, including helix unwinding and catalytic core formation, with free energy calculations.

In our simulations, we observed the instability of the APP TMD helix in the C-terminus, which responds sensitively to the hydrated environment and unwinds. The unwinding of the APP TMD in the intracellular pocket of PS1 is thermodynamically spontaneous. At the intracellular interface of PS1, TM2, TM3, and TM5 constitute a solvent-accessible environment to promote α-helix unwinding. The unwinding of the APP TMD is explicitly coupled with its binding to PS1, and the stepwise unwound residues gradually interact with the GVKLG and the PAL motifs of PS1. The formation of a typical anti-parallel β sheet between the unwound APP TMD and the GVKLG motif drives the complex into a stable conformational state for subsequent cleavage. The dynamic and thermodynamic properties of the APP-PS1 complex suggest that the GVKLG motif of PS1 likely acts as an anchor to stably hold the unwound APP TMD, and the PAL motif likely serves as a switch to drive the cleavage site close to the catalytic sites for subsequent proteolysis. The simulations of the Aβ49 TMD-bound PS1 system provide thermodynamic evidence that an anti-parallel β sheet formed between the GVKLG motif and the unwound region is common, which is necessary but not sufficient to deliver the successive sites into the active site of PS1.

In addition to their importance in the unwinding process, water molecules are also involved in the active configurations of the catalytic core of the APP-PS1 complex by forming a water bridge to control the distances between the two catalytic aspartic acids. In the monoprotonated state, the scissile bond T32-L33 or L33-V34 of the APP TMD can slide down and up to interact with the side chains of D257 and D385. Our free energy profiles of the catalytic core in the wild-type system revealed that the L33-V34 scissile bond is more easily hijacked by the two aspartic acids than the T32-L33 scissile bond. The probability of structural configurations formed for Aβ49 cleavage is higher than that for Aβ48 cleavage. Our simulations of the FAD mutant T32N APP system indicated that the mutation can significantly change the free energy landscape, decrease the energy barrier of the sliding of scissile bonds T32-L33 or L33-V34, and increase the hydrogen bond occupancy for Aβ48 cleavage, thus altering the Aβ42/Aβ40 production ratio. We expect similar simulation protocols might also be useful to understand how Aβ42/Aβ40 alteration can be caused by FAD mutations on PS1[45].

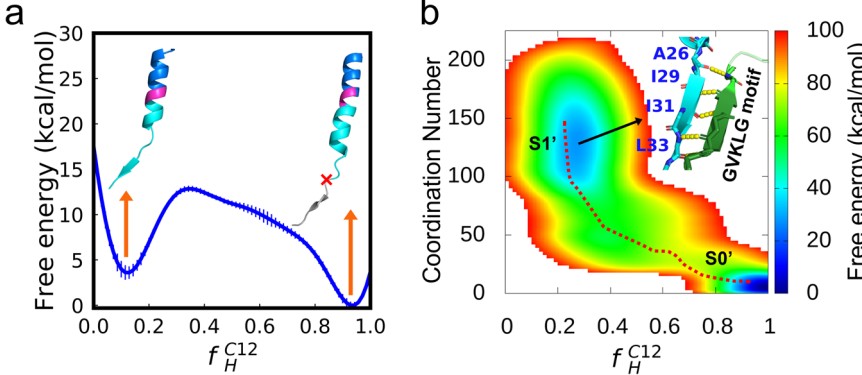

**Fig. 7 Unwinding and binding are necessary for successive delivery of cleavage sites on the APP TMD into the catalytic site of PS1. a** Free energy profile of unwinding as a function of residual helicity for the C12 region of the Aβ49 TMD. Uncertainty is estimated with block analysis. **b** 2D free energy profile as a function of residual helicity for the C12 region of the Aβ49 TMD and the binding states between PS1 and the Aβ49 TMD. The local minimums in the Aβ49 recognition process are labeled S0' and S1', and interactions between the GVKLG motif and the -IVIT- region in state S1' are represented by yellow dashed lines.

In summary, advanced MD simulations have provided insights into the essential steps of APP substrate processing by γ-secretase. In this work CV-based metadynamics methods are used for enhanced sampling, while Gaussian accelerated molecular dynamics have also been used to provide insight into the processing mechanisms of γ-secretase[24,46]. We expect other non-CV-based enhanced sampling methods, such as the milestoning[47] or the weighted ensemble approaches[48], could also be useful for studying the γ-secretase system.

## Methods

**Systems setup and simulation protocol**. All the simulation systems were set up with the CHARMM-GUI protocol[49]. Both the solvent systems and the membrane-solvent systems were set up with TIP3P water and POPC bilayer environments and neutralized with 0.15 M KCl (Fig. 1 and Supplementary Table 1)[50]. Using the CHARMM36m force field[51] and OpenMM package[52], all the simulation systems were minimized and gradually equilibrated. After equilibration, unbiased MD simulations were performed on each system for 200 ns at 1 atm pressure and 310 K with periodic boundary conditions. The long-range interactions were evaluated using particle–mesh Ewald summation with a 12-Å cutoff in real space[53]. The Lennard–Jones interactions were truncated at 12 Å with an atom-based force switching function starting at 10 Å. A time step of 2 fs was used in all the simulations. The bonds involving hydrogen atoms were constrained using the SHAKE algorithm[54]. The snapshots were saved every 100 ps. BE-MetaD simulations were carried out using the Gromacs-2018.6 package[55] patched with the PLUMED 2.5 plugin[56]. The long-range interactions and the Lennard–Jones interactions were calculated using the same parameters in unbiased MD simulations. The bonds involving hydrogen atoms were constrained using the LINCS algorithm[57]. A Nosé–Hoover chain thermostat was used to maintain the temperature at 310 K[58]. A semiisotropic Parrinello–Rahman barostat with a reference pressure of 1 atm and an isothermal compressibility of $4.5 \times 10^{-5}$ bar$^{-1}$ were used to maintain the pressure of the system[59]. In the BE-MetaD simulations, each replica was run for 100 ns, with the bias potential added every 4 ps. The exchange attempt was performed every 10 ps.

**Details on initial structure preparation**. The initial structures of the α-helix fold state APP TMD and APP-PS1 complex were taken from the solution NMR structure (PDB ID: 2LLM) and cryo-EM structure (PDB ID: 6IYC), respectively. In the complex, the double disulfide bonds cross-linking PS1 and APP were removed from PDB 6IYC, and the wild-type residue D385 was restored for computation. For the APP mutant simulation system, T32 was mutated to N32 with local conformational optimization. In the system assuming that the proteolytic reaction occurs at the L33-V34 scissile bond of the APP TMD, the -VMLKKK region on the APP TMD was removed and capped with the neutral C-terminal group. For the initial protonation state of two aspartic acids at the catalytic core, three initial protonation states of D257/D385 were set up, including the inactive state with D257⁻/D385⁻ and the active states with D257H/D385⁻ or D257⁻/D385H.

**Bias-exchange metadynamics**. BE-MetaD is an enhanced sampling technique in which multiple replicas of the simulation system are run in parallel. Except for one neutral replica that is not biased, other replicas are biased over the defined collective variable (CV) using a well-tempered approach (WT-MetaD)[60]. Swaps of

configurations between neighboring replicas are attempted and accepted or rejected after a fixed time interval. In this study, to describe the supposed process of APP cleaved by γ-secretase for the production of a series of Aβs, we used multiple CVs, as listed in Supplementary Table 2, to describe the unwinding state of the substrate TMD and the binding state between the substrate TMD and PS1. Here, we briefly describe the three important CVs: residual helicity, native contact, and coordination number.

The residual helicity is defined according to the following equations:

$$S = \sum_\alpha n \left[ RMSD \left( \{R_i\}_{i \in \Omega_\alpha}, \{R^0\} \right) \right] \qquad (1)$$

$$n(RMSD) = \frac{1 - (RMSD/0.1)^8}{1 - (RMSD/0.1)^{12}} \qquad (2)$$

$$f_H^{C12} = \frac{S}{S_{Max}} \qquad (3)$$

where S is the AlphaRMSD of the target structure, $\{R_i\}_{i \in \Omega_\alpha}$ is the atomic coordinates of a set $_\alpha$, n is a function switching smoothly between 0 and 1, and $\{R^0\}$ are the corresponding atomic positions of the ideal α-helix block. $S_{Max}$ is the maximum value when all selected residues are in the ideal secondary structure (e.g., $S_{Max} = 7$ for 12 residues in the ideal α-helical state). Thus, $f_H^{C12}$ is approximately the percentage of the content of the ideal α-helix block in a peptide[61,62] and is used to characterize the unwound state of the APP TMD and determine the free energy profiles. Likewise, with reference to the structure of the ideal β-sheet block, anti-parallel β sheets can be characterized as AntiBetaRMSD.

The native contact[63] is defined as

$$Q(X) = \frac{1}{N} \sum_{(i,j) \in N} \frac{1}{1 + \exp \left[ \beta \left( r_{ij}(X) - \lambda r_{ij}^0 \right) \right]} \qquad (4)$$

where $Q(X)$ represents the native contacts of conformation X, N is the number of contact pairs of the target structure, and $r_{ij}^0$ is the equilibrium distance between atom i and atom j in the unbiased MD simulation. The pairwise heavy atoms in the target structure with $r_{ij}^0$ smaller than 4.5 Å are selected. β is the smoothing parameter, with $\beta = 50 \, nm^{-1}$. λ was set to 1.5. The unwound state of the APP TMD and the binding state between the APP TMD and PS1 can be described using native contacts.

The coordination number between two groups is defined as

$$CN = \sum_{i \in A} \sum_{j \in B} \frac{1 - \left( \frac{r_{ij}}{r_0} \right)^n}{1 - \left( \frac{r_{ij}}{r_0} \right)^m} \qquad (5)$$

where $CN$ is the coordination number between Group A and Group B, A and B are heavy atoms of two selected regions, the distance $r_0$ between a pair of atoms in contact is set to 4.5 Å, and n and m are the exponents used in the smooth function and set as the default value. To describe the binding state between PS1 and the APP TMD, $CN$ runs over all the pairwise contacts counted from the cryo-EM structure (PDB ID: 6IYC). For the configurations of the catalytic core, $CN$ runs over the heavy atoms between the cleavage sites on the APP TMD and the side chains of D257/D385.

**Simulation analysis**. VMD[64] and MDAnalysis[65] were used for the analysis of the unbiased MD and BE-MetaD trajectories, including geometry calculations, root-mean-square deviations (RMSDs), and root-mean-square fluctuations (RMSFs).

The hydrogen bond is defined with a donor-acceptor distance cut-off of 3.2 Å and an angle cut-off of 30°. The PLUMED-GUI[66] and METAGUI 3[67] plugins were used to analyze the BE-MetaD simulations. The free energy profiles and conformations in the minimums were constructed and analyzed with PLUMED-GUI. The sampled conformations were subjected RMSD-based clustering analysis using METAGUI with a cutoff of 2 Å. To assess the quality of sampling in the well-tempered bias-exchange metadynamics, we examined the time evolution of CVs to confirm that they propagated across relevant ranges during 100 ns MetaD simulations (Supplementary Fig. 5). For free energy profiles, error estimation was performed with block analysis (block size = 100). All structural figures were generated using PyMOL (http://pymol.org/).

**Statistics and reproducibility.** All unbiased MD simulations with difference initial configurations of systems were performed with three replicas. Each simulation used the different random seeds to generate the initial velocity. No data was excluded from the sampled conformations and no blinding methods were used in simulation analysis. The block analysis was applied to characterize the variability of the free energy profiles for the reliability of the results. Statistical analysis was performed using NumPy.

**Reporting summary.** Further information on research design is available in the Nature Research Reporting Summary linked to this article.

## Data availability

The initial data and GROMACS/PLUMED input files required in MD simulations to reproduce the results reported in this paper are available on PLUMED-NEST (www.plumed-nest.org), the public repository of the PLUMED consortium, as plumID:22.000. Source data corresponding to the figures are provided in the supplementary data.

## Code availability

All software used for this study are available and the links are provided as follows. For simulation systems setup: CHARMM c42b1 (https://www.charmm.org/archive/charmm/program/). For MD simulations: OpenMM 7.5.0 (https://simtk.org/projects/openmm), Gromacs-2018.6 package (ftp://ftp.gromacs.org/pub/gromacs/gromacs-2018.6.tar.gz), PLUMED 2.5 plugin (https://github.com/plumed/plumed2). For data analysis: VMD (http://www.ks.uiuc.edu/Research/vmd/), PyMOL (https://pymol.org/2/), METAGUI3 (https://github.com/metagui/metagui3), PLUMED-GUI (https://github.com/giorginolab/vmd_plumed), MDAnalysis package (https://www.mdanalysis.org/).

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

## Acknowledgements

We thank Drs. Zhi Yue and You Xu for valuable advice. The work is supported by the Zhejiang Provincial Natural Science Foundation of China (Grant No. LR19B030001), the National Natural Science Foundation of China (Grant No. 32171247, 21803057), and the Westlake Education Foundation. We thank the Westlake University Supercomputer Center for computational resources and related assistance.

## Author contributions

J.H. designed the experiment, analyzed the data, and wrote the paper. X.L. designed the experiment, performed the simulation, analyzed the data, and wrote the paper.

## Competing interests

The authors declare no competing interests.
