## [Peer Review File · Communications Biology]

Reviewers' comments:

Reviewer #1 (Remarks to the Author):

Comments on the manuscript overall:

In this manuscript, the authors use atomistic molecular dynamics simulations to elucidate the mechanism of APP processing by the presenilin 1 subunit of gamma-secretase. Specifically, they considered the transmembrane domain of APP. There is evidence from past studies that mutations in this region alter the processing of APP by gamma secretase, but the molecular details underlying the difference between mutants were not well understood. In this study, the authors observed a stepwise unwinding of the alpha-helical APP-TMD when bound to presenilin — clearly demonstrating the coupling between a significant conformational change (unwinding of the helix) and binding to PS1 — an exciting result that gives fundamental biophysical insight into this biologically-relevant system. Throughout the manuscript, the authors connect their simulation observations to experimental data for the system, which helps to validate the findings from the simulations and their proposed mechanism. The authors also carried out simulations of the T32N mutation. This mutation was found to significantly alter the dynamics, specifically affecting the 2D free energy profile of coordination number for Aβ48/Aβ49. The findings described by Lu and Huang will be of significant interest to the field because without simulation, understanding the molecular basis for mutations through experiment alone is incredibly challenging. The fundamental insight provided by this work into the processing of APP by gamma-secretase could help in developing novel drugs targeting this important protein-protein interaction, and I would recommend publishing this work with minor revisions (suggested below):

Comments on Statistical analysis:

- Error bars are missing from Fig. 2A. Without an estimate of the error, it's not possible to meaningfully compare the free energy profiles.
- Similarly, error estimates should be provided for Fig. 4A, 6D and S2C.
- While the authors have made use of bias-exchange metadynamics to enhance the conformational sampling in their simulations, it would be helpful if analysis of the quality of sampling was provided. How did the authors determine that the metadynamics simulations were sufficiently long in this case?
- In Table S1, a list of simulation systems and setups are provided. Only one unbiased MD simulation was carried out for the last system (Aβ49-PS1). Is one simulation providing sufficient sampling for this one system and not the others? The authors could comment on this point.

Comments on Reproducibility:

- Overall, the methods section is very clear and detailed. Additional details could be provided in the Simulation Analysis section for reproducibility (i.e. which softwares were used for which analysis, which required in-house code). Some references are missing from the simulation methods section (i.e. for SHAKE, the Nose-Hoover thermostat, the Parrinello-Rahman barostat, TIP3P).

Other comments:

- Fig. 1B, C99 is not defined in the manuscript. References should be added to the scheme for this process.
- page 5, line 134 "In water, the *fHC12* of the APP TMD vibrated violently" - consider revising vibrated violently for clarity.

Reviewer #2 (Remarks to the Author):

In this work, Lu et al. investigated the molecular mechanism of the cleavage of APP TMDs by the γ -secretase using metadynamics and traditional molecular dynamics. Overall, I found the analyses carried out thoroughly and carefully, and the manuscript well written. The findings are very interesting and insightful. The main limitation of this work, in my opinion, is the lack of parallel simulations or error estimations. With that being said, the authors are clearly aware of the limited sampling that the MD simulation can provide, and therefore the choice of the enhanced sampling

method, i.e., metadynamics. Nonetheless, it might be useful for the readers if the authors can acknowledge this limitation in the discussion section. Here are my detailed comments to the authors:

1. The free energy profiles of APP TMD unwinding provided interesting and valuable insights into the differences in the thermodynamic states when the system is exposed to different environments, and the authors also confirmed the existence of these states by observing similar behaviors from unbiased simulations. However, I am wondering if it is feasible for authors to start parallel simulations from different initial conformations, especially for the PS-bound simulation in Figure S1.

2. The authors examined C γ -C γ distance between D257 and D385 using traditional MD simulations. The results indeed suggest that the two residues tend to be closer in the monoprotonated states than in the inactive unprotonated state. However, I wonder how reliable the result is if someone were to run a new simulation with a different starting conformation for the unprotonated state? Also, it seems that the monoprotonated simulations started at a conformation with a closer C γ -C γ distance between D257 and D385 than the unprotonated one did (Figure 5A). I assume that the authors did not interrogate this distance using metadynamics due to biased sampling, but have the authors considered unbiased enhanced sampling methods, such as milestoning or weighted ensemble? This may warrant a discussion in the manuscript. Finally, in line 250, it might be worth reporting an error in the observed distances.

3. What has been thought to be the functional significance of the GVKLG and PAL motif in previous studies? Some explanations may help readers appreciate the findings better if they are not familiar with this particular biological system.

Reviewer #3 (Remarks to the Author):

Lu et al has presented an interesting thermodynamic modeling of the cleavage of transmembrane domain (TMD) of the amyloid precursor protein (APP) by the g-secretase composed of presenilin 1 (PS1) and other three components. The authors performed unbiased molecular dynamics and bias-exchange metadynamics simulations, with focus on g-secretase active site at residue D257 and D385 and substrate APP TMD cleavage sites corresponding to Abeta 48 and 49. To understand its clinical implication, the authors used a familial AD mutation (T to N mutated residue corresponding the C-terminus of Abeta 48) to demonstrate its prediction of increased Abeta 42 to Abeta 40 ratio. The manuscript was clearly written, and the data is well presented.

Some minor points could be addressed in the revised manuscript.

1, Abstract: the significance of this study is not related to the statement that "g-secretase is an important therapeutic target for Alzheimer's disease". While the statement could be controversial, the main goal of this work is to understand the unique endoproteolytic cleavage of the transmembrane domain. In this regard, this study does not address the "recognition of the endogenous substrate APP by g-secretase". Other g-secretase component such as nicastrin may play a more important role in substrate recognition but was not examined in this study.

2, Shi's group from the same institution has published very nice structural data on g-secretase (Nature 2015). To be accurate, Figure 1 C should use available information and provide more detailed illustration of the enzyme-substrate interaction, which is the foundation for MD modeling.

3, Line 203, the authors stated that in the cryogenic condition, the conformational landscape of cryo-EM structure may be different from the one under physiological condition. This is an over-generalized statement to explain outcomes.

4, The authors introduced FAD mutation (T32N) for the modeling. Since there are multiple mutations that enhance Ab42/Ab40 ratio, did the authors substitute other FAD mutated residues in APP? Did the authors obtain similar prediction of the increase in Ab48/49 cleavages?

5, One article by Heilig et al (ref 35) was cited in the manuscript. The same first author also published a study on mutated PS1/g-secretase, (PMCID: PMC3724549), which provides a comprehensive results of Ab42/40 ratios in the presence of different FAD mutations in the g-secretase. In the Discussion section, the authors need to discuss the potential application of their MD modeling for the g-secretase, with the prediction of Ab42/40 changes.

We would like to thank all reviewers for their careful readings and insightful comments. We are very glad to work with them to improve the manuscript.

Reviewer #1

Comments on the manuscript overall:

In this manuscript, the authors use atomistic molecular dynamics simulations to elucidate the mechanism of APP processing by the presenilin 1 subunit of gamma-secretase. Specifically, they considered the transmembrane domain of APP. There is evidence from past studies that mutations in this region alter the processing of APP by gamma secretase, but the molecular details underlying the difference between mutants were not well understood. In this study, the authors observed a stepwise unwinding of the alpha-helical APP-TMD when bound to presenilin — clearly demonstrating the coupling between a significant conformational change (unwinding of the helix) and binding to PS1 — an exciting result that gives fundamental biophysical insight into this biologically-relevant system. Throughout the manuscript, the authors connect their simulation observations to experimental data for the system, which helps to validate the findings from the simulations and their proposed mechanism. The authors also carried out simulations of the T32N mutation. This mutation was found to significantly alter the dynamics, specifically affecting the 2D free energy profile of coordination number for Abeta48/Abeta49. The findings described by Lu and Huang will be of significant interest to the field because without simulation, understanding the molecular basis for mutations through experiment alone is incredibly challenging. The fundamental insight provided by this work into the processing of APP by gamma-secretase could help in developing novel drugs targeting this important protein-protein interaction, and I would recommend publishing this work with minor revisions (suggested below):

Response:

We thank the reviewer for his or her careful readings and favorable comments.

Comments on Statistical analysis:

(1) Error bars are missing from Fig. 2A. Without an estimate of the error, it's not possible to meaningfully compare the free energy profiles.

(2) Similarly, error estimates should be provided for Fig. 4A, 6D and S2C.

Response:

Thank you for pointing this out. Error estimation has been performed with block analysis and Figure 2A, 4A, 6D, 7A, and S2C have been updated accordingly. The following texts have been added in the Method section.

“For free energy profiles, error estimation was performed with block analysis (block size = 100).” (Page 21, line 505)

Figure 2. (A) Free energy profiles of unwinding as a function of residual helicity for the C12 region of the APP TMD, corresponding to APP TMD in membrane (orange), in water (green), and in the PS1 domain (blue). (B) Structural bundles of APP TMD in three simulation systems superimposed onto the middle of the APP TMD. Cyan: C12 region, last 12 residues on the C-terminal helix of APP TMD; magenta: Gly-Gly hinge in APP TMD. f_H^{C12} : the residual helicity of the C12 region.

Figure 4. (A) The distribution of the C_γ - C_γ distance between catalytic sites D257 and D385 in unbiased MD simulations. (B) The stable configurations of D257/D385 are identified according to the C_γ - C_γ distance, including an inactive unprotonated state and three monoprotonated states where the side chains of the two aspartic acid residues are either directly hydrogen-bonded or water-bridged.

Figure 6. (A-C) 2D free energy profiles of coordination formation for Aβ48 and Aβ49 cleavage from BE-MetaD simulations, as well as representative conformations corresponding to free energy minimums. (A) Inactive wild-type protein without protonation (D257⁻/D385⁻), (B) active wild-type protein with protonated D257 and unprotonated D385 (D257^H/D385⁻), (C) active T32N mutant with D257^H/D385⁻. (D) The occupancy of the hydrogen bonds between scissile bonds T32-L33 or L33-V34 and D257/D385 from the unbiased MD simulation for the wild-type and T32N mutant APP systems.

Figure 7. (A) Free energy profile of unwinding as a function of residual helicity for the C12 region of the Aβ49 TMD. (B) 2D free energy profile as a function of residual helicity for the C12 region

of the A β 49 TMD and the binding states between PS1 and the A β 49 TMD. The local minimums in the A β 49 recognition process are labeled S0' and S1', and interactions between the GVKLG motif and the -IVIT- region in state S1' are represented by yellow dashed lines.

Figure S2. (A). The four sampled intermediate states in the APP TMD (blue) are recognized by PS1 (green) corresponding to S0, S1, S2, S3, and S4. (B) Structural comparison of the PS1 response to the APP-bound state (PDB id: 6IYC) and apo-state (PDB id: 5A63), with the conformational difference of the PAL-containing loop in the two structures highlighted. (C) The occupancy of the anti-parallel β sheet formation between the unwound APP TMD and the GVKLG or PAL motif in 200 ns unbiased MD simulations.

(3) While the authors have made use of bias-exchange metadynamics to enhance the conformational sampling in their simulations, it would be helpful if analysis of the quality of sampling was provided. How did the authors determine that the metadynamics simulations were sufficiently long in this case?

Response:

Thank you for this comment. The following sentences are added in the Method section.

“To assess the quality of sampling in the well-tempered bias-exchange metadynamics, we examined the time evolution of CVs to confirm that they propagated across relevant ranges during 100 ns MetaD simulations (Figure S5). For free energy profiles, error estimation was performed with block analysis (block size = 100).” (Page 21, line 503)

Figure S5. Time evolution of the CVs to describe the unwinding state of the substrate TMD and the binding state between the substrate TMD and PS1. (A) Residual helicity of the C12 region of the APP TMD in four simulation systems. (B) Native contact, coordination number, anti-parallel β sheet formation between the unwound APP TMD and the GVKLG and PAL motif in the simulation system of APP TMD bound to PS1. (C) Coordination number between PS1 and scissile bonds in T32-L33 (for A β 48 cleavage) or L33-V34 (for A β 49 cleavage) at the catalytic core in the simulation systems of wildtype APP TMD bound to PS1 and T32N mutant APP TMD bound to PS1. (D) Coordination number, anti-parallel β sheet formation between the unwound A β 49 TMD and the GVKLG and PAL motif in the simulation system of A β 49 TMD bound to PS1.

(4) In Table S1, a list of simulation systems and setups are provided. Only one unbiased MD simulation was carried out for the last system (A β 49-PS1). Is one simulation providing sufficient sampling for this one system and not the others? The authors could comment on this point.

Response:

Thank you for pointing this out. We run two additional replicas for this system and found the results to be consistent among the three replicas, as shown in the new SI

figure (Fig. S1E).

Figure S1. (A-E) Time evolution of fractional residual helicity f_H^{C12} for the C12 region of the APP TMD or A β 49 TMD during unbiased MD simulations with three replicas under different conditions. (F) The C12 region of the APP TMD is surrounded by water molecules located at the intracellular hydrophilic pocket (electrostatic potential surface) in PS1. The deepest position to which water molecules can penetrate into PS1 is near the V28-I29-V30 region of the APP TMD.

Comments on Reproducibility:

(5) Overall, the methods section is very clear and detailed. Additional details could be provided in the Simulation Analysis section for reproducibility (i.e. which softwares were used for which analysis, which required in-house code).

Response:

Thank you for your suggestions. We add the details of the simulation analysis in the Methods section.

“Simulation Analysis. VMD¹ and MDAnalysis² were used for the analysis of the unbiased MD and BE-MetaD trajectories, including geometry calculations, root-mean-square deviations (RMSDs), and root-mean-square fluctuations (RMSFs). The hydrogen bond is defined with a donor-acceptor distance cut-off of 3.2 Å and an angle cut-off of 30°. The PLUMED-GUI³ and METAGUI³ plugins were used to analyze the BE-MetaD simulations. The free energy profiles and conformations in the minimums were constructed and analyzed with PLUMED-GUI. The sampled conformations were subjected RMSD-based clustering analysis using METAGUI with a cutoff of 2 Å. To assess the quality of sampling in the well-tempered bias-exchange metadynamics, we examined the time evolution of CVs to confirm that they propagated across relevant ranges during 100 ns MetaD simulations (Figure S5). For free energy profiles, error estimation was performed with block analysis (block size = 100). All structural figures were generated using PyMOL (<http://pymol.org/>).” (Page 19)

(6) Some references are missing from the simulation methods section (i.e. for SHAKE, the Nose-Hoover thermostat, the Parrinello-Rahman barostat, TIP3P).

Response:

Thank you for the comment. We add the corresponding references in the Methods section (Ref. ⁵⁻¹⁰, which are Refs. 50, 53, 54, 57, 58, and 59 in the main file).

Other comments:

(7) Fig. 1B, C99 is not defined in the manuscript. References should be added to the scheme for this process.

Response:

Thank you for the comment. We add the definition of C99 in the Introduction section.

“Human γ -secretase is a unique and fascinating four-component protease complex that functions in the final cleavage of the transmembrane domain (TMD) of the 99-aa C-terminal fragment of amyloid precursor protein (APP C99).” (Page 2, line29)

(8) page 5, line 134 “In water, the f_{HC12} of the APP TMD vibrated violently” - consider revising vibrated violently for clarity.

Response:

Thank you for pointing this out. We revised it as “In water, the f_H^{C12} of the APP TMD fluctuates significantly”.

Reviewer #2

In this work, Lu et al. investigated the molecular mechanism of the cleavage of APP TMDs by the γ -secretase using metadynamics and traditional molecular dynamics. Overall, I found the analyses carried out thoroughly and carefully, and the manuscript well written. The findings are very interesting and insightful. The main limitation of this work, in my opinion, is the lack of parallel simulations or error estimations. With that being said, the authors are clearly aware of the limited sampling that the MD simulation can provide, and therefore the choice of the enhanced sampling method, i.e., metadynamics. Nonetheless, it might be useful for the readers if the authors can acknowledge this limitation in the discussion section.

Response:

We thank the reviewer for his or her careful readings and favorable comments.

Here are my detailed comments to the authors:

(1) The free energy profiles of APP TMD unwinding provided interesting and valuable insights into the differences in the thermodynamic states when the system is exposed to different environments, and the authors also confirmed the existence of these states by observing similar behaviors from unbiased simulations. However, I am wondering if it is feasible for authors to start parallel simulations from different initial conformations, especially for the PS-bound simulation in Figure S1.

Response:

Thank you for the comment. For each system we perform three parallel unbiased simulations starting from the same initial conformations but with different random seeds. The results are consistent between parallel runs (Figure S1), and are also consistent with the metadynamics simulations. We also analyzed the time evolution of CVs during metadynamics (Figure S5). We hope these two additional figures (S1 and S5) could help illustrate the degree of robustness of our MD results.

(2) The authors examined C γ -C γ distance between D257 and D385 using traditional MD simulations. The results indeed suggest that the two residues tend to be closer in the monoprotonated states than in the inactive unprotonated state. However, I wonder how reliable the result is if someone were to run a new simulation with a different starting conformation for the unprotonated state?

Response:

Thank you for the comment. The starting C γ -C γ distances between D257 and D385 are similar for the monoprotonated and the inactive unprotonated states, however during the equilibration the C γ -C γ distance quickly shortened to about 4.5 Å as shown in the Figure below. We thus think that the shorter distance between D257 and D385 might not be due to a particular starting conformation.

Fig. R1. The time evolution of the C γ -C γ distance between catalytic sites D257 and D385 in unbiased MD simulations under two different protonated states. The dashed vertical line separates the equilibration (eq) and production (200 ns) simulation states.

(3) Also, it seems that the monoprotonated simulations started at a conformation with a closer C γ -C γ distance between D257 and D385 than the unprotonated one did (Figure 5A). I assume that the authors did not interrogate this distance using metadynamics due to biased sampling, but have the authors considered unbiased enhanced sampling methods, such as milestoning or weighted ensemble? This may warrant a discussion in the manuscript.

Response:

Thank you for your suggestion. For both monoprotonated and unprotonated simulations, the initial distances between D257 and D385 are very similar while for monoprotonated states the distance quickly shortened during the equilibration as shown in the above Fig. R1. As the transition of the C γ -C γ distance of D257/D385 is captured in unbiased simulations, it is not used to drive the conformational changes in metadynamics. We add the following sentences in the Conclusions section.

“In this work CV-based metadynamics methods are used for enhanced sampling, while Gaussian accelerated molecular dynamics have also been used to provide insight into the processing mechanisms of γ -secretase¹¹. We expect other non-CV-based enhanced sampling methods, such as the milestoning¹² or the weighted ensemble approaches¹³, could also be useful for studying the γ -secretase system.” (Page 17, line 425)

(4) Finally, in line 250, it might be worth reporting an error in the observed distances.

Response:

Thank you for your suggestion. We changed “approximately 4.5 Å (orange curve) and 6.5 Å (green curve)” into “4.5±0.4 Å (orange curve) and 6.5±0.3 Å (green curve)”. (Page 10, line 254)

(5) What has been thought to be the functional significance of the GVKLG and PAL motif in previous studies? Some explanations may help readers appreciate the findings better if they are not familiar with this particular biological system.

Response:

Thank you for the suggestion. We add the following sentences in the Results section.

“In particular, the interaction contributed by the GVKLG motif (located in the loop between TM6 and TM7) and the PAL motif (located in the loop between TM8 and TM9) of PS1 was analyzed. These two motifs were known to be involved in the substrate recognition of γ -secretase for the active conformation¹⁴, as their mutations or deletions can directly alter or even eliminate the substrate cleavage activity^{15,16}.” (Page 7, line 190)

Reviewer #3

Lu et al has presented an interesting thermodynamic modeling of the cleavage of transmembrane domain (TMD) of the amyloid precursor protein (APP) by the γ -secretase composed of presenilin 1 (PS1) and other three components. The authors performed unbiased molecular dynamics and bias-exchange metadynamics simulations, with focus on γ -secretase active site at residue D257 and D385 and substrate APP TMD cleavage sites corresponding to Abeta 48 and 49. To understand its clinical implication, the authors used a familial AD mutation (T to N mutated residue corresponding the C-terminus of Abeta 48) to demonstrate its prediction of increased Abeta 42 to Abeta 40 ratio. The manuscript was clearly written, and the data is well presented.

Response:

We thank the reviewer for his or her careful readings and favorable comments.

Some minor points could be addressed in the revised manuscript.

(1) Abstract: the significance of this study is not related to the statement that “ γ -secretase in an important therapeutic target for Alzheimer’s disease”. While the statement could be controversial, the main goal of this work is to understand the unique endoproteolytic cleavage of the transmembrane domain. In this regard, this study does not address the “recognition of the endogenous substrate APP by γ -secretase”. Other γ -secretase components such as nicastrin may play a more important role in substrate recognition but was not examined in this study.

Response:

Thank you for pointing this out. Indeed, this study focuses on the processing instead of the recognition of APP by γ -secretase. The corresponding sentences in the abstract has been rewritten as follows.

“Human γ -secretase cleaves the transmembrane domains (TMDs) of amyloid precursor protein (APP) into pathologically relevant amyloid- β peptides (A β s). The detailed mechanisms of the unique endoproteolytic cleavage by the presenilin 1 domain (PS1) of γ -secretase are still poorly understood.” (Page 1, line 10)

(2) Shi’s group from the same institution has published very nice structural data on γ -secretase (Nature 2015). To be accurate, Figure 1 C should use available information and provide more detailed illustration of the enzyme-substrate interaction, which is the foundation for MD modeling.

Response:

Thank you for your suggestion. We added a new panel in Figure 1 to illustrate the details of enzyme-substrate interactions in the system.

Figure 1. (A) Sequence overview of A β 49 APP TMD and successive cleavage sites for A β 37-49 production, the 12 C-terminal residues of APP TMD are highlighted in cyan. (B) The proposed process of APP C99 (99-residue fragment of APP) cleavage by γ -secretase for the production of a series of A β s. (C) Simulation systems were used to investigate the dynamic and thermodynamic properties of APP TMD (in blue) recognized by PS1 (in green) of γ -secretase, including APP TMD solvated in the water environment, APP TMD embedded in the membrane environment, APP TMD bound to PS1, and A β 49 TMD bound to PS1. (D) Enzyme-substrate interaction between APP TMD and PS1 of γ -secretase with key residues of PS1 involved in the interaction with unwound APP highlighted. The PAL motif (P433-A434-L435) and two β sheets (β 1 and β 2) from PS1 stabilize the APP β sheet via the hybrid anti- β sheet formation. The mutation A385 for the experimental structure determination is restored as D385 in the simulation systems.

(3) Line 203, the authors stated that in the cryogenic condition, the conformational landscape of cryo-EM structure may be different from the one under physiological condition. This is an over-generalized statement to explain outcomes.

Response:

Thank you for pointing this out. We revised this sentence and cited a recent work of Rukmankesh Mehra et al. as follows.

“The difference might also due to the ‘cryogenic’ condition in the cryo-EM experiments, as the biological entities examined are frozen in vitreous ice¹⁷. A recent simulation study on γ -secretase showed that cryo-temperature can affect the radius of gyration and the β -sheet formation of substrate-bound γ -secretase¹⁸.” (Page 8, line 206)

(4) The authors introduced FAD mutation (T32N) for the modeling. Since there are multiple mutations that enhance Ab42/Ab40 ratio, did the authors substitute other FAD mutated residues in APP? Did the authors obtain similar prediction of the increase in Ab48/49 cleavages?

Response:

Thank you for the comment. The main topic of this work is to investigate the molecular mechanism of APP processing by PS1. We included one set of calculations with the FAD mutation T32N as this mutation is located at the cleavage site T32-L33-V34 region, so it should have direct impact on A β 48/A β 49 cleavage alteration that can be compared with the experimental measurements. As the purpose is to demonstrate that our general understanding of the mechanism is reasonable, and because of the extensive computational costs associated with BE-MetaD simulations, we didn't perform simulations with the other FAD mutations in APP.

(5) One article by Heilig et al (ref 35) was cited in the manuscript. The same first author also published a study on mutated PS1/g-secretase, (PMCID: PMC3724549), which provides a comprehensive results of Ab42/40 ratios in the presence of different FAD mutations in the g-secretase. In the Discussion section, the authors need to discuss the potential application of their MD modeling for the g-secretase, with the prediction of Ab42/40 changes.

Response:

Thank you for this suggestion. We add the following sentence in the Conclusion section.

“We expect similar simulation protocols might also be useful to understand how A β 42/A β 40 alteration can be caused by FAD mutations on PS1¹⁹.” (Page 17, line 421)

References

- 1 Humphrey, W., Dalke, A. & Schulten, K. VMD: visual molecular dynamics. *J Mol Graph* **14**, 33–38, 27–38, doi:10.1016/0263-7855(96)00018-5 (1996).
- 2 Michaud-Agrawal, N., Denning, E. J., Woolf, T. B. & Beckstein, O. MDAAnalysis: a toolkit for the analysis of molecular dynamics simulations. *J Comput Chem* **32**, 2319–2327, doi:10.1002/jcc.21787 (2011).
- 3 Giorgino, T. PLUMED-GUI: An environment for the interactive development of molecular dynamics analysis and biasing scripts. *Comput Phys Commun* **185**, 1109–1114, doi:10.1016/j.cpc.2013.11.019 (2014).
- 4 Giorgino, T., Laio, A. & Rodriguez, A. METAGUI 3: A graphical user interface for choosing the collective variables in molecular dynamics simulations. *Comput Phys Commun* **217**, 204–209, doi:10.1016/j.cpc.2017.04.009 (2017).
- 5 Martyna, G. J., Klein, M. L. & Tuckerman, M. Nose-Hoover Chains – the Canonical Ensemble Via Continuous Dynamics. *J Chem Phys* **97**, 2635–2643, doi:Doi 10.1063/1.463940 (1992).
- 6 Vangunsteren, W. F. & Berendsen, H. J. C. Algorithms for Macromolecular Dynamics and Constraint Dynamics. *Mol Phys* **34**, 1311–1327, doi:Doi 10.1080/00268977700102571 (1977).
- 7 Darden, T., York, D. & Pedersen, L. Particle Mesh Ewald – an N.Log(N) Method for Ewald Sums in Large Systems. *J Chem Phys* **98**, 10089–10092, doi:Doi 10.1063/1.464397 (1993).
- 8 Hess, B., Bekker, H., Berendsen, H. J. C. & Fraaije, J. G. E. M. LINCS: A linear constraint solver for molecular simulations. *Journal of Computational Chemistry* **18**, 1463–1472, doi:Doi 10.1002/(Sici)1096-987x(199709)18:12<1463::Aid-Jcc4>3.0.Co;2-H (1997).
- 9 Jorgensen, W. L., Chandrasekhar, J., Madura, J. D., Impey, R. W. & Klein, M. L. Comparison of Simple Potential Functions for Simulating Liquid Water. *J Chem Phys* **79**, 926–935, doi:Doi 10.1063/1.445869 (1983).
- 10 Martyna, G. J., Tuckerman, M. E., Tobias, D. J. & Klein, M. L. Explicit reversible integrators for extended systems dynamics. *Mol Phys* **87**, 1117–1157, doi:Doi 10.1080/00268979600100761 (1996).
- 11 Bhattarai, A. *et al.* Mechanism of Tripeptide Trimming of Amyloid beta-Peptide 49 by gamma-Secretase. *J Am Chem Soc* **144**, 6215–6226, doi:10.1021/jacs.1c10533 (2022).
- 12 Tang, Z. Y., Chen, S. H. & Chang, C. E. A. Transient States and Barriers from Molecular Simulations and the Milestoning Theory: Kinetics in Ligand-Protein Recognition and Compound Design. *Journal of Chemical Theory and Computation* **16**, 1882–1895, doi:10.1021/acs.jctc.9b01153 (2020).
- 13 Suarez, E. *et al.* Simultaneous Computation of Dynamical and Equilibrium Information Using a Weighted Ensemble of Trajectories. *Journal of Chemical Theory and Computation* **10**, 2658–2667, doi:10.1021/ct401065r (2014).
- 14 Wang, J., Brunkan, A. L., Hecimovic, S., Walker, E. & Goate, A. Conserved “PAL” sequence in presenilins is essential for gamma-secretase activity, but not required for formation or stabilization of gamma-secretase complexes.

- Neurobiol Dis* **15**, 654-666, doi:10.1016/j.nbd.2003.12.008 (2004).
- 15 Wang, J. *et al.* C-terminal PAL motif of presenilin and presenilin homologues required for normal active site conformation. *J Neurochem* **96**, 218-227, doi:10.1111/j.1471-4159.2005.03548.x (2006).
- 16 Xiao, X. *et al.* APP, PSEN1, and PSEN2 Variants in Alzheimer's Disease: Systematic Re-evaluation According to ACMG Guidelines. *Front Aging Neurosci* **13**, 695808, doi:10.3389/fnagi.2021.695808 (2021).
- 17 Ourmazd, A. Cryo-EM, XFELs and the structure conundrum in structural biology. *Nat Methods* **16**, 941-944, doi:10.1038/s41592-019-0587-4 (2019).
- 18 Mehra, R., Dehury, B. & Kepp, K. P. Cryo-temperature effects on membrane protein structure and dynamics. *Phys Chem Chem Phys* **22**, 5427-5438, doi:10.1039/c9cp06723j (2020).
- 19 Heilig, E. A., Gutti, U., Tai, T., Shen, J. & Kelleher, R. J., 3rd. Trans-dominant negative effects of pathogenic PSEN1 mutations on gamma-secretase activity and Abeta production. *J Neurosci* **33**, 11606-11617, doi:10.1523/JNEUROSCI.0954-13.2013 (2013).

REVIEWERS' COMMENTS:

Reviewer #1 (Remarks to the Author):

The authors have addressed the comments and concerns raised by all reviewers and revised the manuscript accordingly. I would recommend that the manuscript be published and I have no further concerns or questions.

Reviewer #2 (Remarks to the Author):

I would like to thank the authors for the thorough revision. I just have one more comment for the authors to consider for future work. This is regarding my original comment #1 and the authors' rebuttal, I understand the stochastic nature of the three parallel simulations that the authors performed, but for obtaining a complete free energy profile, typically multiple simulations with different starting points are required, as the free energy landscape can be greatly biased by the initial states and their surrounding energy landscape. However, considering the reproducibility of the states from the three parallel simulations and the low dimensional (1D) free energy profile, I find the revision acceptable.

Reviewer #3 (Remarks to the Author):

The authors have addressed my multiple points; no further revision is needed.